# Chromosome length and gene density contribute to micronuclear membrane stability

Anna E Mammel[1] , Heather Z Huang[1] , Amanda L Gunn[1] , Emma Choo[1], Emily M Hatch[1,2]

**Micronuclei are derived from missegregated chromosomes and frequently lose membrane integrity, leading to DNA damage, innate immune activation, and metastatic signaling. Here, we demonstrate that two characteristics of the trapped chromosome, length and gene density, are key contributors to micronuclei membrane stability and determine the timing of micronucleus rupture. We demonstrate that these results are not due to chromosome-specific differences in spindle position or initial protein recruitment during post-mitotic nuclear envelope assembly. Micronucleus size strongly correlates with lamin B1 levels and nuclear pore density in intact micronuclei, but, unexpectedly, lamin B1 levels do not completely predict nuclear lamina organization or membrane stability. Instead, small gene-dense micronuclei have decreased nuclear lamina gaps compared to large micronuclei, despite very low levels of lamin B1. Our data strongly suggest that nuclear envelope composition defects previously correlated with membrane rupture only partly explain membrane stability in micronuclei. We propose that an unknown factor linked to gene density has a separate function that inhibits the appearance of nuclear lamina gaps and delays membrane rupture until late in the cell cycle.**

## Introduction

Micronuclei (MN) in metazoans form around chromosomes or chromosome fragments that missegregate during mitosis and recruit their own nuclear envelope (NE). MN are biomarkers of chromosome instability in cancer, frequently arise during early embryogenesis in humans, and occur at a low frequency in healthy tissue (Guo et al, 2018). Similar to nuclei, MN are enclosed by a nuclear membrane and typically have a nuclear lamina and nuclear pore complexes (NPCs), although often at a reduced density (Hatch et al, 2013; Liu et al, 2018). MN can support major nuclear functions, including transcription, DNA replication, and DNA damage repair, although these can be attenuated or delayed (Terradas et al, 2009, 2012; Crasta et al, 2012; Hatch et al, 2013; Liu et al, 2018). However, persistent rupture of the MN

membrane, which causes loss of MN compartmentalization for the duration of interphase, is frequent in cultured cells, cancer tissue, and early embryogenesis (Hatch et al, 2013; Vázquez-Diez et al, 2016; Liu et al, 2018; Daughtry et al, 2019; Willan et al, 2019). MN rupture arrests micronuclear functions and leads to aneuploidy, DNA damage, and activation of innate immune and cell invasion pathways (Hatch et al, 2013; Zhang et al, 2015; Ly et al, 2016; Harding et al, 2017; Mackenzie et al, 2017; Bakhoum et al, 2018; Soto et al, 2018; Mohr et al, 2021). DNA damage caused by MN rupture is thought to be a major driver of chromothripsis and kataegis, two "all-at-once" processes that cause chromosomal rearrangements and hypermutations, respectively (Stephens et al, 2011; Nik-Zainal et al, 2012). A current model for chromothripsis is that MN rupture causes fragmentation of the encapsulated chromatin, which remains together through mitosis and is re-ligated by error-prone DNA damage repair pathways upon incorporation into a nucleus or MN in the next cell cycle (Crasta et al, 2012; Hatch et al, 2013; Zhang et al, 2015; Ly et al, 2016, 2019; Umbreit et al, 2020; Leibowitz et al, 2021). Despite the high frequency of MN rupture and its potential to drastically change gene expression, the molecular mechanisms of membrane rupture in MN and its full consequences are unclear.

MN frequently have large gaps in the nuclear lamina meshwork, leaving areas of weak membrane that become the site of membrane rupture (Hatch et al, 2013). Similar gaps enable and are the site of membrane rupture in the nucleus (Maciejowski & Hatch, 2020). These gaps lack multiple major NE components, including NPCs, lamins (B-type and, frequently, A-type), and most NE transmembrane proteins (Maciejowski & Hatch, 2020). Lamin B1 depletion is sufficient to cause nuclear lamina gap formation in nuclei (Lenz-Böhme et al, 1997; Vergnes et al, 2004; Shimi et al, 2008; Hatch & Hetzer, 2016) and impaired lamin B1 recruitment is thought to underlie defects in nuclear lamina organization and membrane stability in MN as well (Okamoto et al, 2012; Hatch et al, 2013; Liu et al, 2018; Kneissig et al, 2019; Xia et al, 2019). Overexpression of lamin B1, or its related protein lamin B2, is sufficient to inhibit nuclear membrane rupture in both MN and nuclei (Vargas et al, 2012; Hatch et al, 2013; Maciejowski et al, 2015; Hatch & Hetzer, 2016; Bakhoum et al, 2018). However, nuclear lamina gaps can form many hours before the MN loses integrity (Hatch et al, 2013), suggesting that other mechanisms trigger membrane rupture. Actomyosin

---

[1]Basic Sciences Division, Fred Hutchinson Cancer Research Center, Seattle, WA, USA  [2]Human Biology Division, Fred Hutchinson Cancer Research Center, Seattle, WA, USA

Correspondence: ehatch@fredhutch.org

compression likely accelerates rupture of very large MN (Liu et al, 2018), but the trigger in most cases is unknown (Hatch & Hetzer, 2016).

A current model for reduced NE protein recruitment to MN is that the microtubule-dense midspindle prevents targeting of critical components, including lamin B1 and NPCs, to lagging chromosomes during NE assembly by inhibiting protein dephosphorylation or physically impairing ER access (Afonso et al, 2014; Karg et al, 2015; Castro et al, 2017; Liu et al, 2018). In these models, chromosomes missegregating outside the spindle or at the spindle poles generate larger MN that recruit near normal amounts of lamin B1 and NPCs and rupture less frequently compared to MN formed around chromosomes in the midspindle (Liu et al, 2018). Other data suggest high membrane curvature and small nuclear size are sufficient to impair lamin B1 meshwork assembly in both the nucleus and MN and cause rupture (Xia et al, 2018, 2019; Kneissig et al, 2019; Pfeifer et al, 2021).

The composition and stability of the nuclear lamina could vary widely between single chromosome MN depending on the identity of the entrapped chromosome. Heterochromatin is a key regulator of nuclear lamina organization, nuclear mechanical stability, and nuclear membrane integrity, and its density varies widely between chromosomes in the human karyotype (Furusawa et al, 2015; Stephens et al, 2018). One type of heterochromatin, called lamina associated domains (LADs) (Pickersgill et al, 2006; Guelen et al, 2008), localizes to nuclear periphery and interacts directly and indirectly with a number of nuclear lamina proteins, including lamin A, LBR, and Lap2B (Pyrpasopoulou et al, 1996; Zullo et al, 2012; Solovei et al, 2013; Hoskins et al, 2021). Differences in chromosome length and centromere size between individual chromosomes could also indirectly affect MN nuclear lamina recruitment by biasing chromosome position to outside or within the midspindle during missegregation (McIntosh & Landis, 1971; Mosgöller et al, 1991; Booth et al, 2016). Thus, chromosome identity could contribute to MN stability through multiple mechanisms.

In this study, we demonstrate that chromosome identity is a major determinant of MN rupture timing and nuclear lamina structure. Analysis of single chromosome MN finds that chromosome length, which correlates with MN size, has an additive effect to gene density on membrane integrity and that both features delay membrane rupture. Chromosome-based MN stability differences are not due to a bias in missegregation positioning. Chromosomes correlated with high and low MN stability have similar midspindle missegregation localization and similar NPC recruitment defects during post-mitotic NE assembly, suggesting that differences occur at a later time point. Instead, we find a strong correlation by early G1 phase between lamin B1 levels, NPC density, and MN size. Surprisingly, small gene-dense MN have very low levels of lamin B1 but are less likely to have nuclear lamina gaps compared with gene-poor MN of similar size, suggesting that gene density is a strong predictor of nuclear lamina organization. Our data confirm a connection between MN size and nuclear lamina composition, but suggest that an intrinsic factor linked to high gene density is sufficient to inhibit nuclear lamina disorganization even in the absence of lamin B1. Together, these results demonstrate that analyzing MN chromosome content will be critical to understand the mechanisms of MN rupture and the cellular consequences of micronucleation in different disease contexts.

# Results

To analyze chromosome-specific differences in MN stability, we first established a robust system to identify single micronucleated chromosomes by FISH, using commercially available *Homo sapiens* (HSA) chromosome specific probes combined with immunofluorescence (IF) against a centromere protein. Single chromosome MN were generated in hTERT-RPE-1 cells, a near-diploid chromosomally stable cell line, by first synchronizing these cells in G1 with a Cdk4/6 inhibitor (PD-0332991; Cdk4/6i) then releasing cells into an Mps1 inhibitor (reversine; Mps1i), which blocks the spindle assembly checkpoint (Santaguida et al, 2010) (Fig 1A). MN rupture frequency was assessed by histone H3K27 acetylation (H3K27ac) IF. Similar to a previously used rupture marker, H3K9ac (Hatch et al, 2013; Mohr et al, 2021), H3K27ac had a strong positive correlation with 3xGFP-NLS (nuclear localization signal) and a strong negative correlation with 2xGFP-NES (nuclear exclusion signal) in MN, both of which are well characterized nuclear integrity markers (Fig S1A–D) (Hatch et al, 2013; Denais et al, 2016; Takaki et al, 2017; Vietri et al, 2020). In addition, only a small proportion of MN were H3K27ac positive and 3xGFP-NLS negative, suggesting that almost all MN were able to import proteins (Fig S1B). Consistent with these results, we also observed a similar decrease in the number of H3K27ac-positive MN during interphase (Fig S1E and F) as that reported using GFP-NLS (Hatch et al, 2013; Zhang et al, 2015; Liu et al, 2018). To validate H3K27ac as an integrity marker for MN containing small gene-poor chromosomes, we assessed H3K27ac labeling of single chromosome HSA 18 MN. HSA 18 MN frequently had reduced H3K27ac labeling compared with the nucleus, but the signal was sufficiently high to distinguish intact from ruptured MN (Fig 1B). This was true for all other chromosomes examined in this study (Fig S1G and H). Together, these data demonstrate that H3K27Ac is a sensitive and accurate marker of MN integrity.

To determine whether chromosome identity correlated with MN stability, and which chromosome features regulated membrane integrity, we examined a panel of 10 chromosomes spanning the distribution of chromosome length (fivefold, [NIH–Human genome assembly GRCh38.p13]), gene density (3.5-fold, [Worrall et al, 2018]), centromere size (4.2-fold, [Miga et al, 2014]), ribosomal DNA (rDNA) presence, and centromere position in the human karyotype (Figs 1C and S1H). Rupture frequency was compared between different single-chromosome MN 24 h post-release into Mps1i (Fig 1A) when ~50% of MN were ruptured (Fig S1F) and cells were in G1/S (Fig S1I and J). We found consistent chromosome-specific differences in MN stability across multiple experimental replicates (Fig 1D), with several chromosomes having a high likelihood of maintaining MN stability throughout G1. Analysis of MN rupture frequency of highly intact chromosome MN at a later time point in S/G2 (Fig S1I and J) found that chromosome identity delays, but does not prevent rupture (Fig 1E).

Examination of the traits most closely correlated with MN stability identified chromosome length and gene density as directly proportional to MN integrity. HSA-1, -11, -20, and -22 have a similar gene density (20–23 genes/Mb) but vary fivefold in length, and the proportion of intact MN (MN stability) consistently increases with increasing length (Fig 1F and H). Similar results are observed for HSA-4, -13, and -18, which have a lower gene density (12–13 genes/

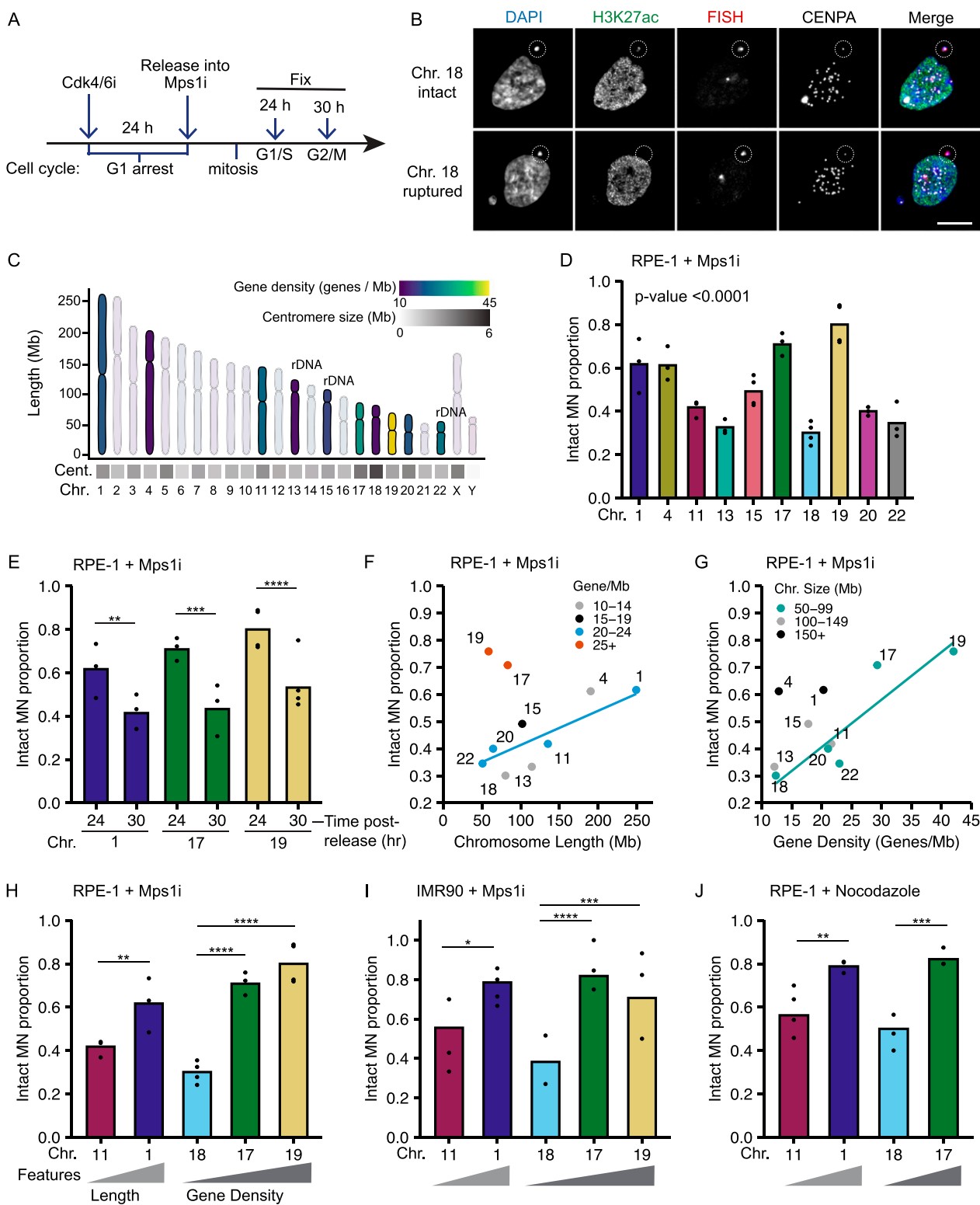

**Figure 1. Micronucleated chromosome length and gene density correlate with membrane rupture.**
**(A)** Time course of IF/FISH experiments. RPE-1 cells arrested in G1 by incubation in PD-0332991 (Cdk4/6i) for 24 h, then released into reversine (Mps1i) to induce MN formation. Cells fixed 24 or 30 h post-Cdk4/6i release, at G1/S and S/G2, respectively (Fig S1E and F). **(B)** Maximum intensity projection images of intact (H3K27ac+) and ruptured (H3K27ac−) MN containing a single HSA 18 (dotted circle). Scale bar = 10 μm. **(C)** Characteristics of selected chromosomes (bold) span a wide range of chromosome length, centromere size and positioning, rDNA repeats, and gene density. Chromosome layout based on Worrall et al (2018). **(D)** Intact proportion 24 h post-Cdk4/6i release for single chromosome containing MN. Chi-square family test; *P*-value < 0.0001; N = (3–4); n = (107, 67, 67, 92, 120, 72, 133, 125, 95, and 84). **(E)** Comparison of MN

Mb) and vary twofold in length (Fig 1F). Surprisingly, comparison of HSA-17, -18, -19, -20, and -22, which have similar length (50–100 Mb) but a 3.4-fold variation in gene density, showed a consistent increase in MN stability with increasing gene density (Fig 1G and H). We expected that increasing LAD density would correlate with increased MN stability, due to increased nuclear lamina interactions and mechanical resistance, but instead we observed a negative correlation between LAD density and MN stability (Fig S2A), consistent with our gene density results. No correlations were observed between centromere size and MN stability (Fig S2B), and MN stability was not altered by the presence of rDNA/acrocentric centromeres when compared to chromosomes of similar length and gene density (Fig S2C and D). To determine whether these correlations between gene density or chromosome length and stability were conserved across MN formation mechanisms and cell lines, we assessed the rupture frequency of HSA-1 versus -11 (chromosome length) and HSA-18 versus -17 and/or -19 (chromosome gene density) after using an alternative Mps1i (BAY-1217389), nocodazole release, which increases MN by increasing prometaphase duration, and in IMR90 fibroblasts treated with Mps1i (reversine). In each case, we found that MN containing larger or more gene dense chromosomes were more stable (Figs 1H–J and S2E), suggesting that these correlations are independent of the type of mitotic defect or RPE-1–specific gene expression.

Chromosome volume in the nucleus is proportional to the length of the chromosome (Eils et al, 1996; Kemeny et al, 2018) and we identified a similar relationship between MN size and chromosome length (Fig 2A). To analyze MN size, we used the maximum projected area, which was strongly correlated with MN volume across the range of sizes observed (Fig S3A). In RPE-1 cells, MN shape was closer to an oblate spheroid than a ball across all sizes. Thus, nearly all MN, even small ones containing HSA-18 and -19, had areas of higher curvature around the edges and flatter curvatures on the top and bottom (Fig 2B and C). To test the hypothesis that MN size determines rupture frequency, we induced multi-chromosome MN by treating with a higher dose of Mps1i (Fig S3B). Both the median MN area and the proportion of intact MN increased with centromere number (Fig 2D and E), consistent with increased size improving stability. However, we found that even very large multi-chromosomal MN were not protected from membrane rupture later in the cell cycle (Fig S3C).

To determine the relationship between MN size and gene density, we assessed MN rupture frequency in MN containing either one or two copies of HSA 18 (gene-poor) or HSA 19 (gene-dense) (Fig 2F–H). For both chromosomes, doubling the number of alleles increased the median area (Fig 2F and G). Increasing MN size rescued the membrane instability of the gene-poor HSA 18 MN and

further increased the stability of the gene-dense HSA 19 MN (Fig 2H). These data indicate that MN size is additive to gene density and suggest that they regulate MN stability through independent mechanisms.

Larger chromosomes tend to segregate on the exterior of the metaphase plate during mitosis (McIntosh & Landis, 1971; Mosgöller et al, 1991; Booth et al, 2016), suggesting that our MN stability results could be an indirect effect of different chromosome missegregation positions. To address this hypothesis, we first assessed the location of missegregating chromosomes with different MN stabilities (HSA-1, -11, -17, and -18) during post-mitotic NE assembly, defined as the time between the first appearance of lamin A on anaphase chromatin and the loss of a broad midspindle region, visualized by labeling with α-tubulin (Fig S4A). Our analysis found no significant difference in chromosome missegregation position regardless of chromosome length, gene density, or MN membrane stability (Figs 3A–C and S4B). Second, we analyzed the stability of single chromosome HSA 1 MN when chromosome missegregation was biased towards the spindle pole, by incubation in a CENPE inhibitor (CENPEi; GSK-923295), or the midspindle, by incubation in nocodazole (Fig S4C and D). After a short arrest, nocodazole cells were released into fresh medium and CENPEi cells were released into Mps1i to inhibit error correction. Our analysis was limited to HSA 1, as that was the only chromosome we observed missegregating at spindle poles after release from CENPEi (Fig S4C and D). Consistent with chromosome identity being a larger determinant of MN stability than missegregation position, we observed no change in HSA 1 MN stability in CENPEi compared with nocodazole (Fig S4E).

Chromosomes missegregating in the midspindle have reduced recruitment of "non-core" proteins, including nucleoporins (Nups), lamin B1, and LBR, during NE assembly (Afonso et al, 2014; Karg et al, 2015; Castro et al, 2017; Liu et al, 2018). If differences in midspindle association are present between chromosomes, we would expect to observe differences in non-core protein recruitment as well. To test this hypothesis, we analyzed recruitment of Nup133, an early nuclear pore assembly protein (Otsuka & Ellenberg, 2018), to HSA-1, -11, -17, and -18 chromosomes missegregating in the midspindle. Nup133 recruitment was substantially reduced on lagging chromosomes compared to the main chromatin mass, consistent with previous results (Liu et al, 2018; Afonso et al, 2019), but no substantial difference was observed for different chromosomes (Fig 3C). Together with our observation that chromosome position is not biased during missegregation, these data strongly suggest that chromosome-specific effects on membrane stability are not due to altered initial protein recruitment during NE assembly.

To determine how chromosome length and gene density affect non-core NE protein levels in interphase MN, we analyzed lamin B1

rupture frequency 30 and 24 h post-release. 30 h; N = (3, 3, and 4); n = (111, 67, 94). **(D)** 24 h; replotting of data from panel (D). **(D, F, G, H)** Replotting of data from panel (D). **(F, G)** MN stability (proportion intact) positively correlates with chromosome length when grouped by gene density, (G) and with gene density, when grouped by chromosome length. Only groups with > 3 chromosomes were analyzed (line). **(H)** MN stability correlates with chromosome length and gene density for representative chromosomes HSA-1 and -11 (length), and HSA-18, -17, and -19 (gene density). **(A, I, J)** MN stability for single chromosome MN in IMR90 cells, treated as in panel (A) (I), and in RPE-1 cells after nocodazole release (J). Correlations between MN stability and chromosome length and gene density were observed in both conditions. IMR90: N = (4, 3, 3, and 3); n = (42, 36, 33, 68, and 58). RPE-1 nocodazole: N = (3, 4, 3, and 3); n = (85, 80, 56, and 56). For all bar graphs in the article, individual experiments are represented as points and pooled replicate proportions are represented as bars. For all sample sizes, N = number of experimental replicates, n = total number of objects/bar. For all bar graphs, chi-square tests are performed for before pairwise comparisons by Barnard's exact test. *P*-values are Barnard's exact test, except where indicated. *P < 0.05, **P < 0.01, ***P < 0.001, ****P < 0.0001. Cent, centromere; Chr, chromosome; Mb, megabase.

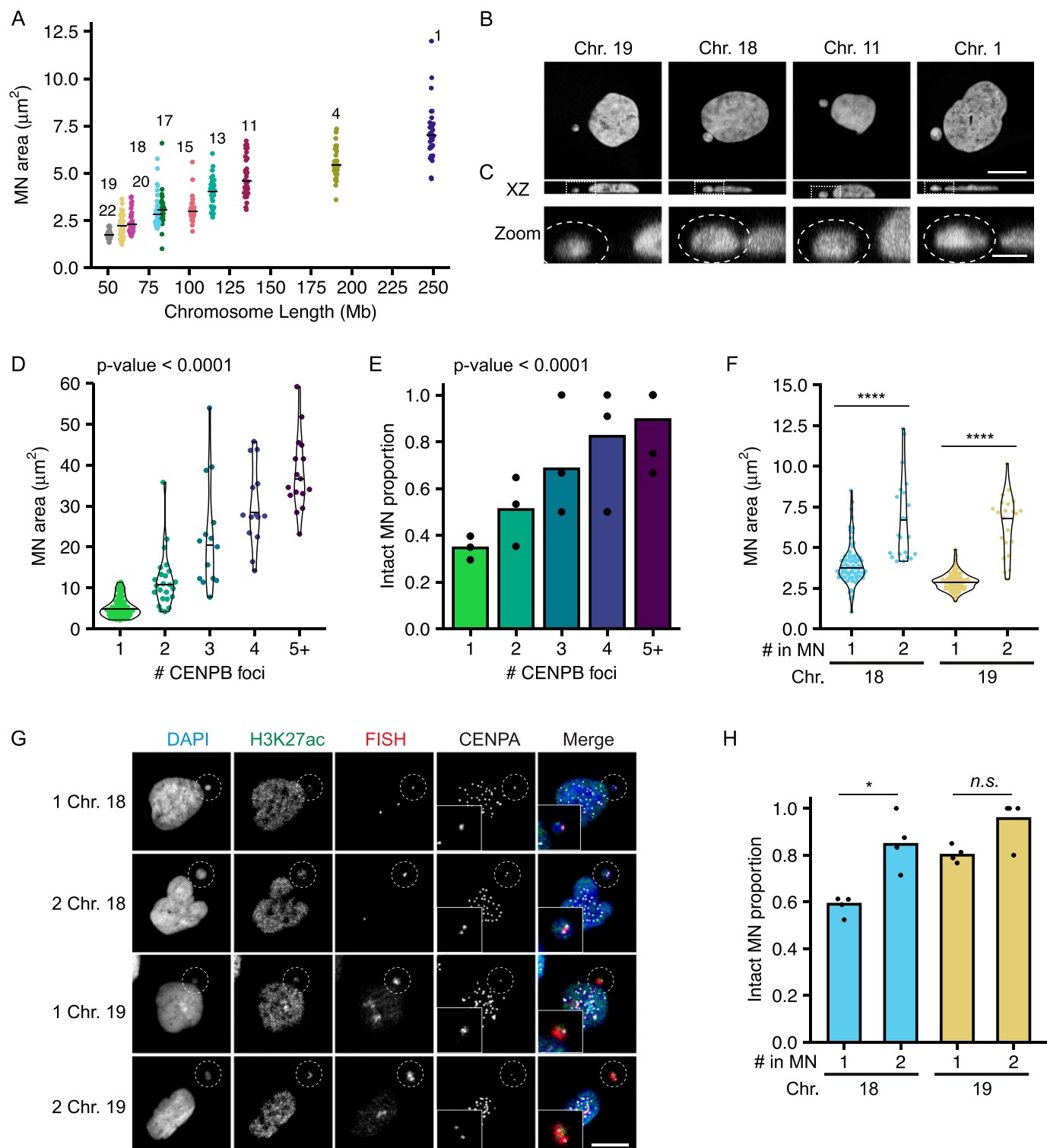

**Figure 2. MN size correlates with MN stability and has an additive effect to gene density.**
**(A)** Maximum projected area of intact MN 24 h post-Cdk4/6i release is correlated with chromosome length. N = (3–4), n = (25, 30, 33, 30, 38, 30, 28, 30, 31, and 30) MN per chromosome. **(B)** Example images of DAPI stained intact single chromosome MN containing indicated chromosomes (FISH/CENPA labeling not shown). Orthogonal sections (XZ planes) shown for each. Scale bar = 10 $\mu$m. **(C)** Zoomed in image of the XZ orthogonal section (white box) containing the MN (outlined). Z-step size = 0.15 $\mu$m; 21 steps; scale bar = 2 $\mu$m. **(D, E)** Quantification of intact MN area (D) and MN stability (E) in RPE-1 cells 24 h post-Cdk4/6i release into 1 $\mu$M MPS1i. MN containing single and multiple chromosomes, determined by CENPB foci number, were analyzed. All MN lacking a centromere were ruptured, and therefore not included in area analysis. **(D)** P-value = Kruskal–Wallis one-way test. N = 3, n = (61, 24, 14, 14, and 17). **(E)** P-value = Chi-square. N = 3, n = (10, 177, 48, 20, 17, and 19). **(F)** MN containing two copies of HSA-18 or -19 are significantly larger than MN containing a single copy. Welch two sample t test; **** P-value < 0.0001; N = 4, n = (72, 22, 92, and 22). **(G)** Images of intact (H3K27ac+)

levels and Nup133 density on MN in early G1 (Fig S1I and J). Lamin B1 intensity was quantified from confocal microscopy images and Nup133 foci were counted using stimulated emission depletion microscopy (STED). This super-resolution imaging technique provides single NPC resolution, with each NPC represented by a single Nup133 focus (Otsuka et al, 2016). We cannot rule out that some Nup133 foci in our images mark immature NPCs (Doucet et al, 2010). However, the presence of GFP-NLS in most H3K27Ac-positive MN (Fig S1A and B) suggests that almost all MN have active NPCs in our system, consistent with the presence of at least one Nup133 focus per MN (Fig 4D). As expected, both lamin B1 protein levels and Nup133 foci density were reduced on MN compared to nuclei (Figs 4A–F and S5A and B) (Hatch et al, 2013; Liu et al, 2018; Kneissig et al, 2019; Xia et al, 2019). We observed that lamin B1 intensity and Nup133 density strongly correlated with MN size, but not with MN stability, especially in small chromosome MN (Fig 4C and F). HSA-17, -18, and -19 MN had similar reductions in lamin B1 intensity and Nup133 foci density (Fig 4B and E), yet HSA-17 and -19 MN were significantly more stable than HSA-18 MN (Fig 1H). To determine whether the correlation between MN size and protein levels was limited to single chromosome MN, we analyzed lamin B1 and Nup133 in larger MN containing multiple chromosomes. We found that lamin B1 and Nup133 recruitment increased in multi-chromosome MN compared with single chromosomes (Figs 4C and F and S5A and B). Furthermore, we observed that individual chromosomes had a large variance in lamin B1 and Nup133 amounts (Fig 4B and E), similar to the variance in MN area observed for specific chromosomes (Fig 2A). These data strongly suggest that MN size is the main determinant of non-core protein levels in interphase and that neither lamin B1 nor NPC amount is sufficient to predict membrane stability.

Because nuclear lamina gaps are a strong predictor of nuclear membrane rupture in multiple systems (Maciejowski & Hatch, 2020), we next asked whether MN stability correlated with nuclear lamina organization. We used lamin A to analyze nuclear lamina organization because it co-localizes with lamin B1 in MN (Hatch et al, 2013), and unlike lamin B1, it is consistently and strongly recruited to lagging chromosomes and MN, enabling visualization of the lamina meshwork even in small MN (Castro et al, 2017; Liu et al, 2018). To assess nuclear lamina morphology in single-chromosome MN, we imaged MN at early G1 using STED. Nuclear lamina gaps were identified by segmenting the lamin A meshwork using an algorithm based on Kittisopikul et al (2020) and classifying them based on size and lamin A intensity (Fig 4G). This analysis was able to identify nuclear lamina gaps in single chromosome and multi-chromosome MN (Fig S5C), indicating its utility across a broad range of MN sizes. Analysis of nuclear lamina gap number identified a consistent trend of more stable MN, including larger or more gene-dense MN, being more likely to have no nuclear lamina gaps compared with more unstable MN (Fig 4H). Analysis of gap sizes and numbers in single-chromosome MN found no substantial difference between MN of different stabilities (Fig S5D and E). A modest decrease in gap size in small compared with large MN was observed (Fig S5D), likely related

to available surface area. These results suggest a model where high gene density or large size delays nuclear lamina gap formation, but does not affect the morphology of the gaps once they form. Together, our data indicate that large MN size and higher lamin protein levels are insufficient to maintain an intact nuclear lamina, and that an unknown function connected to high gene density is critical to maintain meshwork organization and membrane stability.

## Discussion

In this study, we demonstrate that chromosome properties are a critical determinant of MN membrane stability. We identify conserved correlations between membrane stability and increased chromosome length and gene density. MN containing a large chromosome or a small, gene dense chromosome rupture later during interphase compared with small gene-poor chromosomes. These correlations cannot be solely explained by differences in chromosome missegregation position. Instead, we find that chromosome length and chromosome number are directly proportional to MN size and lead to increased levels of lamin B1 and NPC density. High gene density, on the other hand, leads to decreased nuclear lamina gaps, even on small MN depleted of lamin B1 and NPCs. Our data show that MN size has an additive effect with MN stability on gene density, consistent with two independent mechanisms of action. Overall, our data support the existing model that nuclear membrane rupture requires nuclear lamina gaps, but strongly suggest that lamin B1 and NPC depletion are insufficient to explain why MN have more nuclear lamina defects compared with nuclei. Instead, we propose that an additional factor, regulated by gene density, determines the appearance of nuclear lamina gaps and the timing of membrane rupture (Fig 5).

We observe strong correlations between chromosome length, gene density, and MN stability across multiple mitotic disruptions and in multiple cell lines. Interestingly, the proportion of intact MN overall and for a given chromosome differs between these conditions (Fig 1H). Although some of this difference is likely due to differences in cell synchronization efficiency and timing, these observations suggest that chromosome identity acts on top of preexisting conditions in the cell that determine the overall likelihood of MN rupture. This could explain potentially contradictory results in the literature, such as the surprising stability of MN containing the short, gene-poor HSA Y in DLD-1 cells (Ly et al, 2016). DLD-1 MN have an unusually low rupture frequency compared with other cancer cells (26% ruptured MN in DLD-1) (Ly et al, 2016), compared with 45% and 65% in U2OS and DU145 (Hatch et al, 2013), which could reflect an overall delay in MN rupture timing even for high unstable MN. In summary, identifying MN content will be critical for future studies on the mechanisms of MN stability, as biases in the number or identity of micronucleated chromosomes could substantially alter the overall rupture frequency.

MN containing one or two copies of HSA-18 and -19. Chromosome number determined by CENPA foci number. Scale bar = 10 $\mu$m. **(H)** Stability of MN containing one versus two copies of HSA-18 or -19. Two copies increased stability for both chromosomes. Two MN containing a single copy. Barnard's exact test; *$P$ < 0.05, *n.s.* $P$ > 0.05. N = 4, n = (122, 26, 115, and 23).

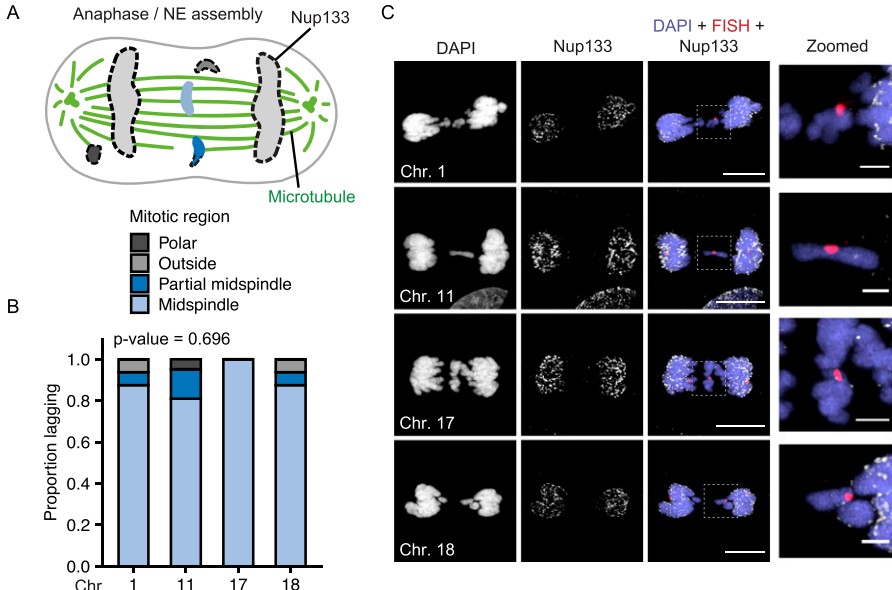

**Figure 3. Post-mitotic missegregation position and non-core protein recruitment are not correlated with chromosome identity.**
**(A)** Model of chromosome missegregation positions and non-core protein (i.e., Nup133) recruitment based on Liu et al (2018). Missegregated chromosomes can be within the midspindle (light blue), partially within the midspindle (blue), outside the midspindle (gray), or polar (dark gray). **(B)** Position of indicated missegregated chromosomes during early to mid-nuclear envelope assembly (see the Materials and Methods section for definitions) in RPE-1 cells 13–15 h post-Cdk4/6i release. *P*-value = Chi-square. N = 3, n = (16, 21, 11, 16). **(C)** Images of Nup133 recruitment to indicated chromosomes during midspindle missegregation. Images are deconvolved single sections (z-step size = 0.15 μm). Scale bar = 10 μm. Zoomed in panels show fewer Nup133 foci on all chromosomes compared with main mass. Scale bar = 3 μm.

In the nucleus, nuclear lamina gaps are frequently located at areas of high membrane curvature and previous analyses have observed correlations between MN rupture timing, lamin B1 levels, and MN size, leading to a hypothesis that high average membrane curvature in MN causes loss of lamin B1, leading to lamina gaps, and increased membrane tension, leading to rupture (Hatch et al, 2013; Thiam et al, 2016; Xia et al, 2018, 2019; Deviri et al, 2019; Kneissig et al, 2019; Pfeifer et al, 2021 *Preprint*). Consistent with this model, we find that larger MN have larger areas of low curvature membrane, more lamin B1, better nuclear lamina organization, and reduced rupture frequency (Fig 5). Whether higher lamin B1 levels in the MN are caused by lower membrane curvature, increased protein import (Liu et al, 2018), or another mechanism acting after NE assembly (Fig 3), is currently unclear. However, we also observe that MN of all sizes have both fairly flat and highly curved regions, that lamina gaps and rupture frequency do not correlate only with MN size, and that lamina gaps are present on both highly curved and relatively flat MN surfaces (Fig 4G). These data suggest that curvature is only one property regulating nuclear lamina organization and membrane stability and that its effects can be suppressed by other factors.

We demonstrate that gene density is an important factor for MN stability, but it is unclear what aspect of gene density regulates nuclear lamina organization. Gene density correlates strongly with GC content, euchromatin histone modifications, transcription, early replication timing, low LAD density, and higher chromatin mobility in interphase (Schneider & Grosschedl, 2007). It is unknown whether functional differences between gene-poor and gene-dense chromosomes are maintained in MN, but it is likely, at least, that gene-dense chromosomes have substantially more contact with the NE in MN compared to nuclei (Cremer and Cremer, 2001, 2010; Shimi et al, 2008). Based on our observation that high gene density does not prevent membrane rupture or lamina gap formation (Figs 1E and 4H), we hypothesize that one or more of these characteristics delays lamina gap formation, likely through a nuclear lamina independent process.

Chromosome size and gene density determine whether the MN is likely to rupture in G1 or after DNA replication initiates (Fig 5) and this could have significant effects on the consequences of MN rupture. Double-stranded DNA breaks (DSBs) in MN are thought to require DNA replication initiation and be the major type of DNA damage in MN rupturing in S and G2 or entering mitosis without rupturing (Crasta et al, 2012; Hatch et al, 2013; Zhang et al, 2015; Umbreit et al, 2020). In contrast, rupture in G1 may promote mainly ssDNA accumulation, due to TREX-1 endonuclease activity (Mohr et al, 2021). Currently, only MN that rupture after S phase have been shown to undergo chromothripsis (Zhang et al, 2015; Ly et al, 2016, 2019; Umbreit et al, 2020), although TREX-1 has been linked to chromothripsis and kataegis in other contexts (Maciejowski et al, 2015, 2020). Rupture timing also likely determines whether whole or partial chromosome aneuploidy is present in daughter cells. MN that rupture in S/G2 phase prematurely terminate DNA replication, leading to partial aneuploidy in the daughter cells, which can be exacerbated by fragment loss during chromothripsis and amplification of circularized fragments (double-minutes) (Stephens et al, 2011; Zhang et al, 2015; Shoshani et al, 2021). MN that rupture in G1 will have whole chromosome aneuploidy by G2 and likely have impaired kinetochore assembly leading to continued chromosome missegregation in the next cell cycle (Hatch et al, 2013; Soto et al, 2018; He et al, 2019). In addition, the duration that chromatin is exposed to the cytoplasm, or the type of DNA damage, could impact whether cGAS-STING activation occurs (Guey et al, 2020; Mohr et al, 2021). It remains to be seen how differences in rupture timing, and chromosome-specific differences in transcription, replication timing, and NE assembly in MN will affect cell proliferation and immune system activation. However, our results demonstrate that identifying the chromosomes that missegregate into MN in different tissues and cancer types will be critical to understanding how MN rupture drives cancer evolution and disease pathogenesis in vivo.

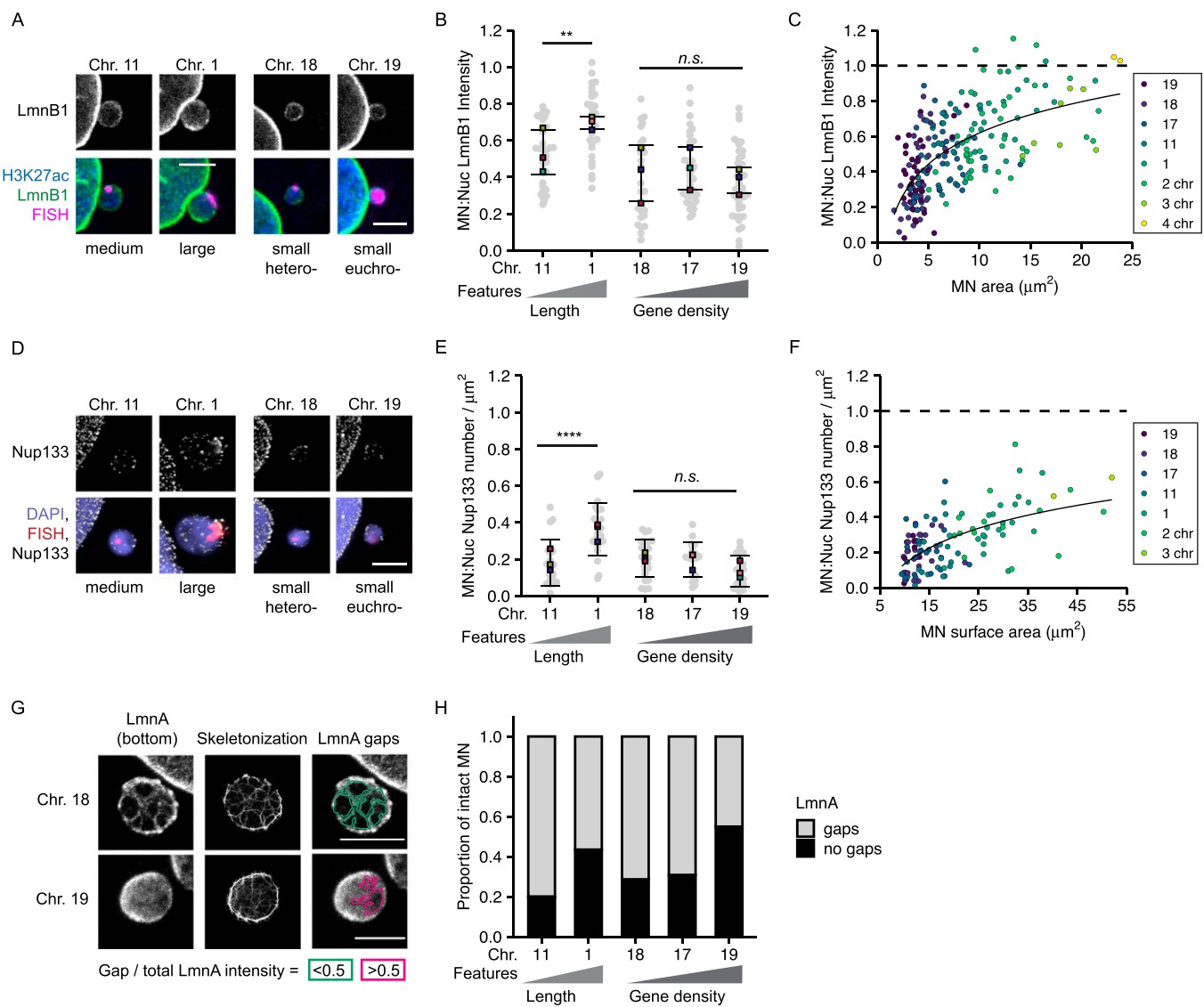

**Figure 4. MN size correlates with nuclear lamina protein levels and gene density inversely correlates with nuclear lamina gaps.**
**(A)** Images of lamin B1 (LmnB1) levels on intact (H3K27ac+) single chromosome MN containing indicated chromosome 20 h post Cdk4/6i release. LmnB1 images are a single section and merged images are maximum-intensity projections. Merge includes H3K27ac (blue) and FISH (magenta). Scale bar = 3 $\mu$m. **(B)** Quantification of MN LmnB1 intensity normalized to nucleus intensity. Lamin B1 intensity correlates with chromosome length, not gene density. Mean from each replicate shown (color squares) with individual measurements (gray circles). One-way ANOVA, $P$-value < 0.0001; pairwise comparison with Bonferroni adjustment, $P$-value < 0.01, n.s. $P$ > 0.05; N = 3, n = (33, 29, 29, 37, and 43). **(C)** LmnB1 intensity correlates with MN area for single and multiple chromosome MN. Chromosome number determined by CREST signal. Dotted line indicates equal MN and nucleus LmnB1 intensity. Spearman's correlation (solid line) r = 0.63, $P$-value < 0.0001. **(B)** N and n for single chromosomes same as panel (B). For multi–chromosome MN, N = 3, n = (41, 9, 2). **(D)** Maximum intensity projections of Nup133 foci on intact single chromosome MN. MN integrity determined by H3K27ac+ signal (not shown). Scale bar = 3 $\mu$m. **(E)** Quantification of Nup133 density (foci number/area) on MN normalized to nucleus density. Mean from each replicate shown (color squares) and individual measurements (gray circles). One-way ANOVA, $P$-value < 0.0001; Bonferroni adjusted pair-wise comparison, *** $P$-value < 0.001, n.s. $P$ > 0.05. N = 3, n = (21, 23, 24, 24, and 21). **(F)** Nup133 density correlates with MN surface area for single and multiple chromosome MN. Spearman's correlation (solid line), r = 0.59, $P$-value < 0.0001. N and n for single chromosomes same as panel (E). For multi chromosomes, N = 3, n = (16, 2). **(G)** Example images of nuclear lamina organization in intact single chromosome MN containing either HSA-18 or -19 labeled with antibodies to lamin A (LmnA). Left = maximum intensity projections of bottom half of z-stack (z-step size = 0.15 $\mu$m). Middle = 3D skeletonization of lamin A structure. Right = detected nuclear lamina gaps (green, pink). Detected gaps were filtered by size (not shown) and intensity. Only gaps where difference between mean fluorescent intensity inside the gap compared with outside was <0.5 (green) were retained. Scale bar = 2 $\mu$m. **(H)** Quantification of nuclear lamina organization in single chromosome intact MN. Chi-square; $P$-value > 0.05. N = 3, n = (25, 23, 21, 26, and 20).

# Materials and Methods

### Cell lines and culture methods

hTERT-RPE-1 (RRID: CVCL_4388) cells were grown in DMEM/F12 (Gibco) + 10% FBS (Gibco) + 1% Pen-Strep (Gibco) + 0.01 mg/ml hygromycin (Sigma-Aldrich) at 5% $CO_2$ and at 37°C. hTERT-RPE-1 3xGFP-NLS is a stable cell line characterized previously (Anderson et al, 2009). IMR90 cells (RRID: CVCL_0347) were grown in DMEM (Gibco) + 15% FBS (Gibco) +1% Pen-Strep (Gibco) at 5% $CO_2$, 5% $O_2$ at 37°C. Cell line identity was determined by short tandem repeat typing.

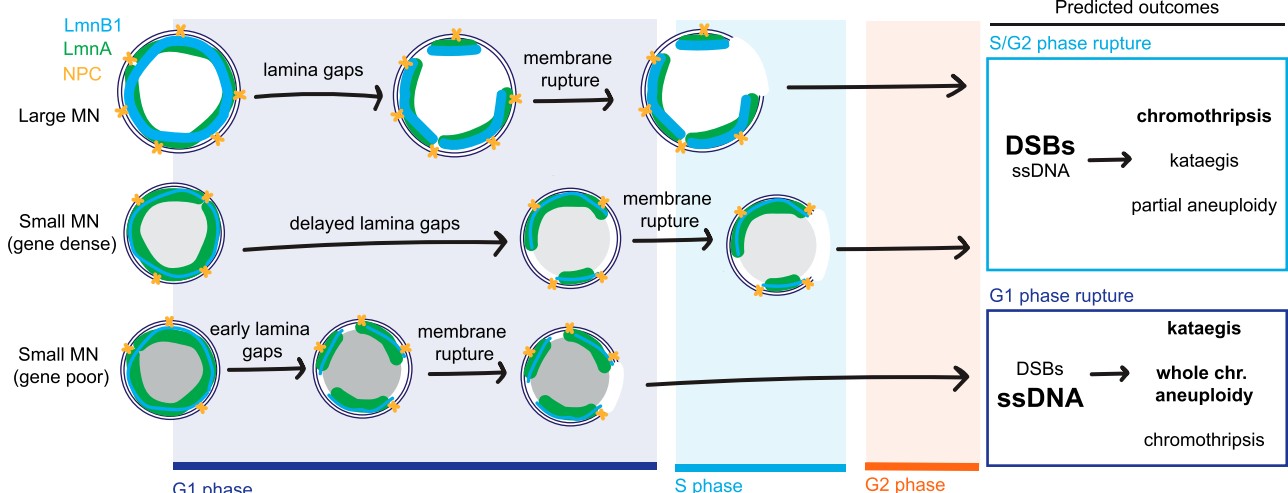

**Figure 5. Size- and gene density–dependent effects on micronuclei lamina structure, rupture timing, and cellular outcomes.**
The nuclear envelope is comprised of an inner and outer membrane, nuclear pore complexes (yellow), and a nuclear lamina containing A and B type lamins (e.g., lamin A [green] and lamin B1 [blue]). Nuclear lamina gaps precede membrane rupture and are inhibited or delayed by increased recruitment of nuclear envelope proteins (large MN) and high gene density (small MN, gene dense). Neither large size nor high gene density is sufficient to prevent membrane rupture, but inhibition of nuclear lamina gap formation delays rupture until S/G2 for most MN in these categories. In contrast, most small, gene-poor MN rupture before DNA replication, in G1. Membrane rupture after DNA replication initiation causes a high frequency of DNA double-strand breaks (DSBs) and chromothripsis. Whereas MN rupture in G1 is strongly associated with accumulation of ssDNA damage, which is associated with kataegis. In the next cell cycle, underreplication and chromothripsis cause partial aneuploidy of the micronucleated chromosome whereas failure to initiate replication leads to whole chromosome loss. Thus, our data demonstrate that chromatin-based factors regulate nuclear lamina organization in MN, and that this could have a significant impact on genome architecture and cell proliferation.

For RPE-1 FISH experiments, except where noted, cells were arrested in G1 by addition of 1 $\mu$M PD-0332991 isethionate (Cdk4/6i; Sigma-Aldrich) for 24 h. Cells were released by washing three times in 1× PBS before incubation in 0.5 $\mu$M reversine (Mps1i; EMD Millipore), 1 $\mu$M where noted, for 14–32 h. The Mps1 inhibitor BAY-1217389 (Thermo Fisher Scientific) was used at 100 nM. For nocodazole treatment and missegregation position experiments, RPE-1 cells were incubated in 100 ng/ml nocodazole (Sigma-Aldrich) or 50 nM GSK-923295 (CENPEi; Cayman Chemicals) for 4–6 h prior to release by washing three times with 1× PBS and then adding either media alone (nocodazole) or media + 0.5 $\mu$M reversine (GSK-923295). For mitotic shake-off experiments, cells were shaken off after last 1× PBS wash and fixed either 45 min (anaphase) or 8 h (G1) post shake-off. IMR90 cells were incubated for 32 h in 1 $\mu$M PD-0332991 isethionate then released into 0.5 $\mu$M reversine for 24 h.

RPE-1 cells were transfected with 2xGFP-NES by nucleofection using a 4D nucleofector (Lonza) and the SE cell line 4D-Nucleofector X Kit S (Lonza). 200,000 cells were resuspended in buffer SE plus 400 ng plasmid, transferred to a 16-well electroporation cuvette, and electroporated using program DS-138. Cells were analyzed 48 h after transfection.

### Plasmids

pDEST53:NES-GFP (2xGFP-NES) was constructed using the Gateway system (Invitrogen) to insert NES-GFP into the N-terminal cycle3-GFP vector, pcDNA6.2/DEST53. The NES sequence is LQLPPLERLTL, from the HIV-1 protein Rev.

### Immunofluorescence (IF) and FISH

Cells were grown on poly-L-lysine coated coverslips and fixed in 4% PFA (Electron Microscopy Sciences) for 10 min at RT for the following

experiments: Mps1i total MN rupture frequency time course, MN area to volume analysis, experiments using hTERT-RPE-1 3xGFP-NLS, 2xGFP-NES, and multi-chromosome MN analysis using a PNA CENPB-Cy5 probe. For all other IF and FISH experiments, cells were fixed in 100% methanol at −20° C for 5–10 min. Coverslips were blocked in 3% BSA (Sigma-Aldrich) + 0.1%–0.4% Triton X-100 (Sigma-Aldrich) + 0.02% sodium azide (Sigma-Aldrich) in 1× PBS (PBS-BT) for 30 min before incubation in primary antibodies diluted in PBS-BT. Primary antibodies used: mouse $\alpha$-tubulin (1:250; 3873S; Cell Signaling Technology), human anti-CREST (1:100; 15-234; Antibodies Incorporated), mouse anti-CENPA (1:100; GTX13939; GeneTex), mouse or rabbit anti-H3K27ac (1:250; 39085, Active Motif; 1:1,000; ab4729; Abcam), rabbit anti-Lamin B1 (1:100; sc-365214; Santa Cruz Biotechnology), rabbit anti-Lamin A (1:500; L1293; Sigma-Aldrich), and rabbit anti-Nup133 (1:100; ab155990; Abcam). Coverslips were washed three times in PBS-BT then incubated in the following secondary antibodies: Alexa Fluor 405-conjugated goat anti-rabbit (1:2,000; A-31556; Thermo Fisher Scientific), Alexa Fluor 488–conjugated goat anti–mouse (1:2000; A-11029; Thermo Fisher Scientific), Alexa Fluor 488–conjugated goat anti–rabbit (1:2,000; A-11034; Thermo Fisher Scientific), Alexa Fluor 594–conjugated donkey anti-rabbit (1:500; 711-585-152; Jackson ImmunoResearch), Alexa Fluor 647–conjugated goat anti–mouse (1:1,000; A-21236; Thermo Fisher Scientific), and Alexa Fluor 647–conjugated goat anti-human (1:1,000; A-21445; Thermo Fisher Scientific). Secondary antibodies were dilute in PBST and incubated for 30 min at RT. Coverslips were washed twice in PBST then incubated with DAPI (1 $\mu$g/ml in PBS; Roche) for 5 min at RT, washed once in diH$_2$O, and mounted in Vectashield (Vector Labs) or Prolong Gold (Life Technologies).

For experiments using chromosome enumeration (XCE) or whole chromosome paint (XCP) probes, after methanol fixation and

immunofluorescence, as described above up until DAPI labeling, coverslips were fixed for 5 min with 4% PFA in 1× PBS. This and subsequent steps were performed at RT unless noted. Coverslips were washed twice with 2× SSC (Sigma-Aldrich) for 5 min then permeabilized with 0.2 M HCl + 0.7% Triton X-100 for 10–15 min at RT. Coverslips were washed twice with 2× SSC for 5 min, denatured in 50% formamide (EMD Millipore) 2× SSC for 1 h, washed twice with 2× SSC, then inverted onto 3–5 µl of Spectrum Orange XCE or XCP probe (MetaSystems) and sealed with rubber cement. Probes and targets were co-denatured at 74°C for 2 min and hybridized 2 h to overnight at 37°C in a humidified chamber. Coverslips were washed once in pre-heated 0.4× SSC buffer at 74°C for 5 min then twice in 4× SSC or 2× SSC + 0.1% Tween-20 for 5 min. Coverslips were incubated in DAPI and mounted in Vectashield (Vector Labs) (for analysis of MN morphology or protein recruitment) or Prolong Gold (Life Technologies).

For FISH with the PNA CENPB-Cy5 probe (PNA Bio), the same protocol was followed until formamide denaturation. At that point cells were incubated in 50% formamide in 2× SSC for 30 min at 85°C then rinsed three times in ice cold 2× SSC. PNA probes were diluted to 50 µM in 85°C hybridization buffer (60% formamide + 20 mM Tris, pH 7.4, + 0.1 µg/ml salmon sperm DNA [Trevigen]) and coverslips were simultaneously washed at 85°C in 2xSSC. Coverslips were then incubated in 10 µl of the PNA probe for 10 min at 85°C and then 2 h at RT. Coverslips were then washed twice with 2× SSC + 0.1% Tween-20 at 55°C for 10 min and once with 2× SSC + 0.1% Tween-20 at RT. before incubation in DAPI and mounting in Prolong Gold (Life Technologies).

## LAD percentage quantification

LAD areas were determined from asynchronous attached RPE1-hTERT cell pA-DamID data available from the 4DN data portal for lamin B2 antibodies (Schaik et al, 2020). Chromosome lengths were determined from GRCh38.p13 human reference genome (GenBank: GCA_000001405.28). ENCODE blacklist regions (Amemiya et al, 2019) were subtracted from LAD data (Schaik et al, 2020) and chromosome lengths before analysis. The LAD percentage per chromosome was determined by dividing the total LAD length (bp) for each chromosome by the chromosome length (bp).

## EdU-pulse labeling and FACS

For FACS cell cycle analysis, cells were incubated with 10 µM EdU (Life Technologies) for 15 min in media before trypsinization and fixation in 70% ice cold ethanol. Cells were stored at –20°C before staining. Fixed cells were washed twice in 1× PBS + 0.1% Triton-X 100 before resuspension in Click-It EdU reaction mix (Alexa-555; Thermo Fisher Scientific) for 30 min while rotating. Cells were washed twice in 1× PBS + 1% BSA and incubated in 1 µg/ml DAPI in 1× PBS for 30 min before analysis. Samples were analyzed on either a three-laser FACSCanto II (BD Biosciences) or a four-laser LSR II (BD Biosciences) and data acquired using DIVA software (BD Biosciences). DNA content was analyzed based on DAPI fluorescence (PacBlue-A), and DNA replication was analyzed based on Alexa-555 fluorescence (PE-A). Doublet discrimination was used to remove doublets and clumped cells using DAPI-A/DAPI-W measurements. Data were

analyzed using FlowJo v.10 software (BD Biosciences). Cell cycle distributions were determined by gating EdU positive versus negative, as determined by single color control, and by 2N versus 4N DAPI content.

## Microscopy

Unless noted below, confocal images were acquired with a Leica DMi8 laser scanning confocal microscope using the Leica Application Suite (LAS X) software and a Leica ACS APO 40×/1.15 Oil CS objective or a Leica ACS APO 63×/1.3 Oil CS objective. Z-stacks were acquired with the system optimized step size except where noted. Confocal images of mitotic cells in Figs 3C and D and S4B were acquired with a Leica TCS SP8 confocal microscope with a Leica HCX Plan Apo 63×/1.40 Oil CS2 objective with a pixel size between 60 and 80 nm and a z-step size of 0.15 µm. Post-acquisition, images were deconvolved using lightning with smoothing a size of medium through the LAS X software. Images for quantification in Fig S1C and D were acquired using a 40×/1.3 Plan Apo objective on an automated Leica DMi8 microscope outfitted with a Yokogawa CSU spinning disk unit, Andor Borealis illumination and an ASI automated stage with Piezo Z-axis. Images were captured with an Andor iXon Ultra 888 EMCCD camera using MetaMorph software (version 7.10.4; Molecular Devices).

Nup133 and lamin A IF labeled cells were imaged using Leica TCS SP8 with the super-resolution microscope system (STED) using a 775 nm pulsed laser, Leica Application Suite software platform (LAS X version 3.5.7.23225), and a Leica HC PL APO 100×/1.4 Oil CS2 objective. Before image acquisition the STED and confocal beams were manually aligned using FluoSpheres mounted in Prolong Gold and white light laser set to 594 and 775 nm STED, the alignment was adjusted until the STED FluoSpheres overlapped with the center of the confocal FluoSpheres images. Images were acquired at ~20 nm pixel size for a resolution of ~50 nm in the xy plane, and a white light laser was tuned to 405 nm (DAPI), 488 nm (H3K27ac), 556 nm (FISH), 594 nm (Nup133 or lamin A), and 647 nm (CREST) wavelengths.

For all images, post-acquisition image processing was limited to cropping the image and adjusting levels through Adobe Photoshop to make use of the entire histogram spectrum. False colors for channels were changed through the arrange channels function in Fiji (Schindelin et al, 2012).

## Image quantification

An MN was defined as a DAPI positive round object adjacent to or near the nucleus that was distinct from the nucleus, to distinguish them from nuclear herniations and chromatin bridge fragments. Teardrop shaped objects were excluded from analysis. Intact MN were defined as those with H3K27ac mean intensity that was equivalent to that of the main nucleus over some part of its area. Ruptured MN were defined as those where the average H3K27ac signal was decreased by >60% compared to the main nucleus. Chromosome number was defined as the number of centromere foci, which were assessed by CENPA or CREST IF, or PNA CENPB-Cy5 FISH. A positive FISH signal was defined as a focus twice the background signal that partially co-localized with a centromere. Interphase cells with more than three FISH foci for a given

chromosome were excluded from analysis as being either tetra-ploid or exceeding acceptable signal to noise ratios.

MN area was calculated from maximum intensity projections by selecting the DAPI channel object and measuring the area in Adobe Photoshop. Pixel area was converted to $\mu m^2$ using the image dimensions.

Missegregated chromosomes in the mitotic positioning analysis were defined as FISH positive chromosomes that were not con-tiguous with the main chromatin mass during early to mid-nuclear envelope assembly. This stage was defined by the presence of a wide spindle midzone and recruitment of lamin A to the main chromatin mass in either a punctate or continuous pattern. These conditions were chosen based on data (not shown) demonstrating that nuclear import starts after the initiation of cytokinesis in RPE-1 cells.

Lamin B1 intensity was quantified for intact (H3K27ac+) MN containing a single CREST focus that overlapped with the FISH probe 20 h post-Cdk4/6i release into MPS1i. A single z-slice was analyzed from the middle of the MN and corresponding nucleus and the average intensity of the entire rim was taken for each MN to minimize the effect of lamina gaps. Images were imported into Adobe Photoshop and the quick selection tool was used to outline the nuclear perimeter of the MN and nucleus from the H3K27ac channel. This selection was converted to a four-pixel border around the rim and the area ($A_I$) and fluorescence integrated density ($F_I$) of the lamin B1 signal in this selection was measured. This selection was then expanded by at least 2× and a second group of area and integrated density fluorescence measurements were taken ($A_O$ and $F_O$). Background subtracted fluorescent intensity values were obtained using the following formula: $F_I - ((F_O-F_I)/(A_I/A_O)) = F_{-N}$.

Nup133 density was determined for intact (H3K27ac+) MN con-taining a single chromosome 20 h post-Cdk4/6i release into Mps1i and its corresponding nucleus. The number of Nup133 foci per nucleus was quantified in Imaris ×64 8.4.2 (Bitplane) by first defining the region of interest around each MN and nucleus using the contour tool, then creating spots for each region of interest with a XY spot diameter set to 0.2 $\mu M$. The threshold was adjusted for each image to capture every Nup133 focus in the nucleus but very few spots in the cytoplasm. The same threshold was used for the corresponding MN. The surface area was calculated in Imaris from the DAPI channel, with smoothing set to 0.5, and background subtraction of 0.2, and the threshold adjusted to encompass the entire DAPI signal.

## Lamin A gap quantification

Nuclear lamina gaps were quantified in intact MN with a single chromosome labeled with lamin A antibodies. A 3D response-weighted segmentation of the lamin meshwork was created in MATLAB (R2020b) using the Adaptive Resolution of Multi-Orientation Space algorithm from Kittisopikul et al (2020), which combines the use of steerable filters with non-maximum sup-pression to identify the center of the lines. The segmented meshwork was divided into two hemispheres (MN top and MN bottom) after identifying the equatorial plane using Brenner's best focus measure method (Pertuz et al, 2013). Top and bottom binary

meshworks were 2D projected and the properties of lamin A gaps (i.e., area, eccentricity, solidity, perimeter, and mean fluorescent intensity) were quantified. "Normal" gaps were filtered out by two criteria, area and intensity, and the thresholds were determined by manual analysis of the lamin A meshwork in nuclei qualitatively defined as having no lamina gaps. The upper limit of manually measured lamin A gaps in nuclei was 0.12 $\mu m^2$, therefore this value was used as a cut-off to define true lamina gaps. Nuclear lamina gaps were also classified based on the ratio of the mean lamin A intensity within an individual gap and the mean intensity of lamin A outside of gaps. For nuclei, this ratio never fell below 0.5 and therefore this value was used as a cut-off. MN were identified as containing a gap if at least one gap was present after filtering. Associated scripts are available at GitHub (https://github.com/hatch-lab/mammel_et_al_2021).

MN volume was calculated in these experiments by taking the average of the inner and outer lamin A volume, determined by a fitted ellipsoid spanning the lamin A–positive voxels and the volume of the convex hull containing all the lamin A–positive voxels, respectively, in MATLAB. The surface area was derived from the volume measurements.

## Statistics

All statistic tests were conducted using R (version 4.0.0) or MATLAB (version 2020b). For all data comprising three or more groups of observations, a family test (i.e., chi-squared for categorical data, ANOVA or Kruskal–Wallis one-way test for continuous data) was performed first to test the null hypothesis that all the observations were the same. Only data where the family test rejected the null hypothesis were further analyzed by multiple comparison testing. The one-way ANOVA test was used on data with normal distribu-tions (determined by the Shapiro–Wilk test) and the Kruskal–Wallis one-way test was used on data where one or more groups deviated from normality. Pair-wise comparisons on categorical data were analyzed using Barnard's exact test (Barnardextest [version 1.0.0.0], Matlab; "Barnard" package, R) and on continuous data using Bonferroni corrected pair-wise comparisons. Statistical analyses of two-group continuous data were performed using Welch unpaired $t$ tests. Significant association between two binary variables (i.e., H3K27ac and GFP-NLS or GFP-NES) was analyzed using the $\varphi$ mean squared contingency coefficient. Spearman's rank correla-tion coefficient was used to assess monotonic relationships for two variables with non-normal distribution (e.g., MN area to volume). For all tests, $P$-values greater than 0.05 were considered statistically significant. A limitation of the chi-square test is that it is highly sensitive to sample size; therefore, a post hoc analysis was per-formed to determine if our datasets reach a chi-square statistical power of 0.8 based on a given effect and sample size using the packages "esc" and "pwr" in R (version 4.0.0). Post hoc power for lamin A gap proportion (Fig 4H) analysis yielded a statistical power = 0.342 given an effect size d = 0.5215 and N = 3, n = (25, 23, 21, 26, and 20). Chi-square statistical tests cannot be performed on datasets with 0 values and are invalid when multiple outcomes have a value less than five. Thus, this analysis was not performed on experi-ments where this was the case.

# Supplementary Information

# Acknowledgements

EM Hatch, AL Gunn, E Choo, and HZ Huang were supported by the National Institutes of Health grant R35GM124766 and a Rita Allen Foundation Scholars Award. This work was supported by the Cellular Imaging, Bioinformatics, and Flow Cytometry Shared Resources of the Fred Hutch/University of Washington Cancer Consortium (P30 CA015704). AE Mammel was supported by the National Cancer Institute of the National Institutes of Health training grant T32CA009657.

## Author Contributions

AE Mammel: conceptualization, data curation, formal analysis, funding acquisition, validation, investigation, visualization, methodology, and writing—original draft, review, and editing.
HZ Huang: conceptualization, formal analysis, investigation, visualization, and methodology.
AL Gunn: formal analysis, validation, and investigation.
E Choo: formal analysis, validation, and investigation.
EM Hatch: conceptualization, supervision, funding acquisition, investigation, project administration, and writing—review and editing.

## Conflict of Interest Statement

The authors declare that they have no conflict of interest.

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
