## [Reviewer comments · Life Science Alliance]

Life Science Alliance

Chromosome length and gene density contribute to micronuclear membrane stability

Anna Mammel, Heather Huang, Amanda Gunn, Emma Choo, and Emily Hatch

DOI: <https://doi.org/10.26508/lsa.202101210>

Corresponding author(s): *Emily Hatch, Fred Hutchinson Cancer Research Center*

Review Timeline:	Submission Date:	2021-08-23
	Editorial Decision:	2021-08-24
	Revision Received:	2021-10-25
	Editorial Decision:	2021-10-26
	Revision Received:	2021-11-01
	Accepted:	2021-11-03

Scientific Editor: *Eric Sawey, PhD*

Transaction Report:

Please note that the manuscript was reviewed at *Review Commons* and these reports were taken into account in the decision-making process at *Life Science Alliance*.

Review
COMMONS

August 24, 2021

Re: Life Science Alliance manuscript #LSA-2021-01210-T

Dr. Emily Hatch
The Fred Hutchinson Cancer Research Center
Basic Sciences Division
1100 Fairview Ave
B2-152
Seattle, WA 98109

Dear Dr. Hatch,

Thank you for submitting your manuscript entitled "Chromosome length and gene density contribute to micronuclear membrane stability" to Life Science Alliance. We invite you to re-submit the manuscript, revised according to your provided Revision Plan.

The typical timeframe for revisions is three months. Please note that papers are generally considered through only one revision cycle.

Thank you for this interesting contribution to Life Science Alliance. We are looking forward to receiving your revised manuscript.

Sincerely,

B. MANUSCRIPT ORGANIZATION AND FORMATTING:

We thank the reviewers for their insightful and constructive comments and their positive assessment of the quality of our work, the novelty of our findings, and their broad scientific appeal. We have updated the revised manuscript with the experiments and clarifications listed below. These changes are shown in the manuscript text in blue. Minor copy edits and edits to improve consistent data reporting in the figure legends were also performed, but not highlighted.

Comments from Reviewer 1

Reviewer #1 (Evidence, reproducibility and clarity):

This manuscript investigates the factors that dictate the timing with which individual micronuclei (MN) undergo interphase rupture. The authors have painstakingly used in situ hybridization to reveal the identity of the mis-segregated chromosome that becomes the MN. Harnessing the variation in the chromosome(s) that generate MN, they determine that chromosome size (or number) and gene density are additively predictive of the relative stability of MN. The authors find that the level of lamin B1 and nuclear pore complexes (NPCs), two factors previously suggested to differ between nuclei and MN and to contribute to MN stability, do scale with the chromosome size encapsulated in the MN. However, gene density appears to represent a novel, independent, variable. The authors relate gene density to a more uniform network of A-type lamins within MN that are more refractory to rupture.

****Specific Points:****

Reviewer #1, comment 1: The study would be extended (and its impact increased) if perturbations designed to build on the correlations outlined in the manuscript were explored. Examples include experiments designed to answer whether the act of transcription (gene activity rather than gene density) protects from rupture (e.g. what effect do Pol II inhibitors have), or whether driving histone modifications associated with gene activity (e.g. HDAC inhibitors) can promote MN stability for gene-poor chromosomes. Does gene density/gene activity influence chromosome decondensation in a manner that could influence MN stability?

Reviewer #2 – Referee Cross-commenting: As mentioned, I think this is a sound study that provides novel insight. It is my view that it should be left to the authors to decide whether they want to strengthen the study with additional experiments to address some of the open mechanistic questions. I do not think they are essential for publication.

Reviewer #1, cross comment on Reviewer #2: I agree with reviewer 2. My suggestions for further mechanistic study was to provide a framework for what might take the paper to the "next level" in terms of impact, which the authors can choose to address or decline depending on their target journal and the editorial assessment of that journal.

- Response review #1, comment 1, cross comment on Reviewer 2 - significance: We agree with reviewer #1 and #2 that further exploration of which characteristics associated with high gene density impact nuclear lamina organization and micronucleus rupture is the critical next step. We also agree with reviewer #2 that these experiments are beyond the scope of the current study. As reviewer #1 points out, many nuclear processes are associated with increased gene density. So far, we have analyzed micronucleus rupture frequency after inhibiting HDACs, RNA Pol II transcription, and DNA polymerases by small molecule treatment and found no bulk effects on MN stability.

[Figure removed by editorial staff per authors' request].

These results suggest that the mechanism is unlikely to be straightforward and will require analysis of these characteristics at the level of individual chromosome MN, as well as analysis of heterochromatin organization, discussed below. We plan to fully pursue this line of inquiry in the future and thus do not plan to add these preliminary negative results to the current manuscript.

Reviewer #1, comment 2: (1) On the surface, it seems surprising that gene-rich chromosomes would assemble a more robust and/or intact nuclear lamina than gene-poor chromosomes, since the latter would be expected to support a greater extent of chromatin-lamin contacts. This aspect of the manuscript could be more thoroughly investigated. Is there any relationship to the ratio of LAD versus non-LAD regions on the chromosomes investigated (the reproducibility across cell lines suggests broad conservation)? **(2)** Does heterochromatin localize to the periphery of micronuclei? **(3)** Or if, as the loss of H3K27Ac suggests, the chromatin is overly heterochromatinized, another epigenetic tool could be to test if inhibiting histone methylation is protective.

- Response Review #1, comment 2 - part 1: We were surprised by this result as well. We have added an analysis of per chromosome LAD density and their correlation with MN rupture to the manuscript (Fig. S2A; page 6, line 27). These data were derived from a recent and publicly available lamin B2 DamID dataset from RPE-1 cells published by the van Steensel lab (van Schaik et al., 2020) through a collaboration with experts in our Bioinformatics Shared Resource group. As expected, low gene density (e.g., chr. 18) correlates very strongly with high LAD density and higher micronuclei rupture frequency. The per chromosome LAD values we obtained were higher than what we expected (70% vs. 50% reported in Guelen et al., 2008). In discussion with the van Steensel lab, we learned that they have observed that epithelial cell lines, like hTERT-RPE-1s and HCT116, have more chromatin interacting with the lamina compared to fibroblasts and other human cell lines and that our LAD proportions are consistent with what they have observed. Based on their response, and the expected anti-correlation with gene density, we believe that our calling of LAD domains is accurate.

- Response Review #1, comment 2 - part 2: Whether heterochromatin localizes properly to the nucleus periphery in MN is a really interesting question, and we think it merits a substantial analysis in future publication(s). Our hypothesis is that interactions between the nuclear lamina and the heterochromatin are unlikely to be fully maintained in MN, but we are still working on directly addressing this question by analyzing how histone modifications are altered on specific chromosomes in MN compared to nuclei, and how peripheral heterochromatin density may be changing. In general, we do see H3K9me2, a modification highly enriched in LADs, at the periphery of both nuclei and MN (see below), but have not examined this on a per-chromosome basis. We currently plan to save these preliminary data for a future publication.

- Response Review #1, comment 2 - part 3: Our data indicate that H3K27Ac loss only occurs when the micronucleus ruptures (Fig. S1A - D), which is often followed by chromatin compaction through an ESCRT-III dependent mechanism (Hatch et al., 2013; Vietri et al., 2020). We have preliminary experiments demonstrating that preserving H3K27Ac after rupture (via TSA treatment) has no effect on rupture frequency, indicating that it is a consequence rather than a cause of rupture. We plan to examine the effect of inhibiting methyltransferases on micronucleus stability as part of a larger investigation into the mechanism by which gene density increases MN stability (see response to comment #1).

Reviewer #1, comment 3: (1) The discussion and justification for the definition of lamin gaps could be clearer. First, it seems essential to indicate in all cases whether the authors are referring to A-type or B-type lamins since the behaviors are clearly distinct. Can the authors justify why the correlation analysis is with lamin B1 levels but then the gaps are indicated for lamin A/C? **(2)** Given that lamin B1 seems the key factor in MN stability from the group's prior work, might the aberrant lamin A/C network actually be deleterious? That is - does knocking down lamin A/C rescue rather than exacerbate the effect? **(3)** Given that lamin B1 seems the key factor in MN stability from the group's prior work, might the aberrant lamin A/C network actually be deleterious? That is - does knocking down lamin A/C rescue rather than exacerbate

the effect? **(4) Further** - what was the authors' rationale for choosing the threshold of $0.12 \mu\text{m}^2$ for lamina gaps?

- Response Reviewer #1, comment 3 – part 1: We thank the reviewer for pointing out this discrepancy and have updated the manuscript to make clear when we are discussing different lamin types. The A-type and B-type lamins typically overlap quite well when lamina gaps or honeycombs are present in nuclei and MN (Hatch et al., 2013). To clarify this point, we have updated the introductory description of lamina gaps (page 3, line 25).
- Nuclear lamina gaps are assessed using lamin A because the lamin B1 signal is too low in small MN to determine the organization of the lamina. As has been previously noted (Castro et al., 2017; Liu et al., 2018), lamin A is recruited at high levels to all MN regardless of their size (which we also observe, compare MN to nuclei in Fig. 4G). This fact makes lamin A a more reliable marker for nuclear lamina organization. This explanation has been added to the text (page 9, line 21).
- Response Reviewer #1, comment 3 – part 2: We looked at the effect of knocking down each of the three main lamin transcripts on MN stability in an earlier paper (Hatch et al, 2013). We found that, in contrast to depletion of lamin B1, depleting lamin A/C or lamin B2 had no effect on bulk MN rupture frequency, suggesting that lamin A/C organization on its own is not driving membrane rupture. Work from other labs has also shown that depleting lamin A/C does not prevent gaps in the nuclear lamina (e.g. Sullivan et al., 1999), but there is evidence that expressing only lamin A in the cell can exacerbate nucleus rupture (Chen et al., 2018). Investigating MN rupture frequency in a triple lamin knockout cell or a lamin A only cell line would be an interesting future line of inquiry.
- Response Reviewer #1, comment 3 – part 3: From our previous work, and that of others in the nuclear lamina field, we know that gaps in the lamin A/C meshwork in MN and nuclei are a marker for a gap in the entire nuclear lamina structure. These gaps lack lamin B1, NPCs, and almost every other NE protein examined (Hatch et al, 2013; Hatch and Hetzer, 2016; Thanisch et al., 2017; Liu et al., 2018; Maciejowski and Hatch, 2020). We added this information to the description of nuclear lamina gaps in the introduction (page 3, line 25). Consistent with lamin A gaps lacking other NE proteins, we observe gaps in both lamin B1 and Nup133 in large MN. To clarify this point we have also added arrows to indicate gaps in the Nup133 foci network in Fig. S5B:

Chr. 1
Zoom

- Response Reviewer #1, comment 3 – part 4: The following text has been added to the materials and methods (page 17, line 9): “Normal` gaps were filtered out by two criteria, area and intensity, and the thresholds were determined by manual analysis of the lamin A meshwork in nuclei qualitatively defined as having no lamina gaps. The upper limit of manually measured lamin A gaps in nuclei was $0.12 \mu\text{m}^2$, therefore this value was used as a cut-off to define true lamina gaps.”
- To further strengthen our definition of lamina gaps, we worked with the Scientific Imaging and Bioinformatics shared resources to develop an automated image analysis pipeline to quantify and

categorize nuclear lamina gaps. This allowed us to refine our definition of lamina gaps through applying both a size filter and a threshold filter based on measurements from normal nuclear laminae and get a more accurate assessment of MN gap content (Fig. 4G, H).

Using this method allowed us to perform additional analyses, including the total number of gaps and the variance of gap sizes per micronucleus (Fig. S5D, E).

Our interpretation of these new results, added to the results section, is that the timing of nuclear lamina gaps in MN is chromosome dependent, but the appearance of these gaps when they form is similar across different micronucleus sizes, different nuclear pore densities, and different gene densities (page 9, line 24 – page 10, line 6).

Reviewer #1, comment 4: The authors state that “all MN contained at least one Nup133 focus” and that “Nup133 density did not correlate with MN stability but instead with MN size..”. What a single Nup133 focus would functionally indicate is not clear (an aggregate of NPCs)? Given the challenge in discerning single NPCs, do the authors arrive at the same conclusion if they use intensity as for lamin B1? Is the distribution of NPCs influenced by the different chromosomes contained within the MN? Or gene density?

- **Response Reviewer #1, comment 4:** As the reviewer points out, we did not evaluate Nup133 distribution, only average density. We do see uneven distributions of NPCs in larger micronuclei, consistent with the gaps in the lamina meshwork (see Fig. 4D, chr 1, left side, and chr. 18). To clarify this point, the text now reads “...each NPC is represented by a single Nup133 focus” (page 9, line 3). STED super-resolution microscopy provides single pore resolution (Otsuka et al., 2016), and therefore our imaging analysis is able to distinguish between single pores and a cluster of pores (page 9, line 2).
- We have also clarified that we cannot conclude that every Nup133 focus is a fully functional NPC (page 9, line 3). As an early recruited protein to this complex, it can be present in a focus on the nuclear membrane prior to full pore development (Doucet et al., 2010).

- To further strengthen our observation that Nup133 levels correlate with MN size, not stability, we have included additional data for chromosome 19 (Fig. 4D-F):

Comments from Reviewer 2

Reviewer #2 (Evidence, reproducibility and clarity): Mammel et al. explore determinants of membrane integrity in micronuclei (MN). Using a panel of 10 human chromosomes and H3K27ac as a marker of membrane integrity, they compare various chromosome features with integrity. They find correlations with chromosome size, with gene density, and with the number of lamin gaps, but not with the presence of ribosomal genes, centromere size or acrocentric centromeres, nor can they be explained by the mitotic position of chromosomes. The authors also find a stabilizing effect of MN size. Lamin B and Nup133 presence did not correlate with stability but with size.

****Specific points****

Reviewer #2, comment 1: The study hinges on the accurate detection of membrane integrity. The authors use H3K27ac for this purpose but it is not clear why this is an accurate marker of membrane integrity. Although the authors provide a rough characterization of this marker, it would be reassuring to validate it against a second more direct marker such as a cytoplasmic soluble, non-imported marker, at least in a set of control experiments.

- Response Reviewer #2, comment 1:** We agree with the reviewer that loss of histone acetylation after micronucleus rupture is not mechanistically understood. Our assumption is that KDAC activity is substantially higher than KAT activity in the cytoplasm, but we have not directly tested this or identified the specific KDAC deacetylating cytoplasmic histones.
- To address the reviewer's concern, we have validated H3K27Ac loss in MN against another well-characterized nuclear membrane rupture marker: GFP-NES (nuclear export sequence). Mislocalization of this reporter, or similar FP-NES reporters, to the nucleus, has been previously validated as a marker for MN and nucleus integrity loss (Hatch et al., 2013; Denais et al., 2016; Takaki et al., 2017; Vietri et al., 2020). As expected, analysis of GFP-NES and H3K27Ac localization to MN showed a strong anti-correlation ($\phi = -0.85$) (Fig. S1C, D).

The text has also been updated (page 5, line 26) to indicate that a similar histone mark, H3K9Ac, was characterized and used as an MN rupture marker in two other studies (Hatch et al., 2013, Mohr et al., 2021).

Comments from Reviewer 3

Reviewer #3 (Evidence, reproducibility and clarity): In their manuscript, the authors investigate factors governing micronuclear membrane rupture and stability. The authors show that multiple characteristics of human chromosomes affect the stability of micronuclei (MN) by comparing 10 chromosomes with diverse features. Identifying chromosomes by fluorescence in situ hybridization (FISH) and using loss of H3K27Ac as a readout for MN stability, they found that chromosome size and gene density correlate with MN membrane stability, whereas centromere size, rDNA presence and centromere position had no effect. Intriguingly, the authors additionally suggest that lamin B1 levels do not predict MN stability, but it instead coincides with gene density. They further suggest that an unidentified property associated with gene density not only delays membrane rupture until late in the cell cycle, but also inhibits the occurrence of nuclear lamina gaps. This is an exciting manuscript, developing interesting new thoughts on the mechanisms that govern MN stability although some of the claims could be more carefully phrased when preparing a revised version for submission. There is no need for further experiments.

- Response Reviewer #3: We appreciate the reviewer's enthusiasm. We have modified our phrasing based on the reviewer's comments, and we thank the reviewer for pointing our areas that needed clarification.

****Specific Points****

Reviewer #3, comment 1: The manuscript presents a set of well-controlled, beautiful experiments that are of commendable quality and in support of most of the final conclusions. The methods used are described in great detail and allow reproducibility. The experiments include statistical analyses where required.

- Response Reviewer #3, comment 1: We thank the reviewer for this positive assessment of our work.

Reviewer #3, comment 2: Based on the presented data, one can draw the interesting conclusion that gene density may matter for MN stability (Fig. 1G). However, it remains unclear based on which Figure or experiment one can in fact conclude that gene density is literally 'additive' to the role of MN size, as stated on page 6? Based on Figure 2G,H one can only conclude on MN size, but not on an 'additive' role of gene density, as proposed. To address this

question, one would have needed to show that changing density one a particular set of chromosomes (e.g. 1 and 18) is additive to the effect of size. But this is not an important conclusion to provide as there is no need to claim that these effects are additive. Please change wording.

- Response Reviewer #3, comment 2: We appreciate the reviewer for bringing this to our attention and have changed the wording to state that our results suggest an additive effect of size on gene density, but not the other way around (abstract, line 6; page 5, line 5; page 7, line 27; discussion page 10, line 19).

Reviewer #3, comment 3: It is confusing that the authors claim at the end of the manuscript that there is no correlation with lamin B1 levels. On p. 4, it is written; 'We find a strong correlation between lamin B1 levels, NPC density, and MN size in interphase.' Later, it is stated that there is a linear relation between MN size, area and chromosome length. Then it is positively concluded that 'These data indicate that MN size and gene density are additive and suggest that they regulate MN stability through independent mechanisms.', highlighting the role of MN size. And, Figure 4B,C shows a correlation between lamin B1 levels and chromosome length. However, then it is suddenly concluded that: 'Surprisingly, lamin B1 intensity and Nup133 density did not correlate with MN stability, but instead with MN size (Fig. 4C, F). But where is this lack of correlation of MN size with MN stability shown, which was a conclusion beforehand, in Figure 1? Couldn't the final conclusion rather be that there is an underlying correlation with lamin B1 levels but perhaps other factors can dominate - e.g. gene density?

- Response Reviewer #3, comment 3: We agree with the reviewer that this was poorly worded and did not precisely communicate our conclusion. Our data do demonstrate that there is a correlation between MN size, lamin B1 recruitment, and MN stability. But this is only a *partial* correlation that cannot account for the better nuclear lamina organization and higher membrane stability of small gene-dense MN. Overall, our results indicate that nuclear lamina organization has the best, although not a perfect, correlation with MN stability, and that the likelihood of proper lamina organization increases with both MN size and gene density. We have updated the text to include this more precise description of our results (abstract, line 10; page 9, line 8; discussion, paragraph 1).

Reviewer #3, comment 4: In the discussion, the authors should compare their finding to what has been reported by Kneissig et al., (eLIFE, 2019), who concluded: 'We show that the isolated micronuclei lack functional lamin B1 and become prone to envelope rupture, which leads to DNA damage and aberrant replication. The presence of functional lamin B1 partly correlates with micronuclei size, suggesting that the proper assembly of nuclear envelope might be sensitive to membrane curvature.'

- Response Reviewer #3, comment 4: We have added a sentence including these data and stating that they are consistent with our results (page 11, line 13). One confounding factor in the Kneissig et al. analysis is that they did not look at rupture status in their analysis of lamin B1 recruitment to micronuclei. We have preliminary unpublished data that ruptured MN can lose lamin B1 and we and others have noted that MN frequently shrink in size upon rupture (Hatch et al., 2013; Vietri et al., 2020). Thus, it is difficult to parse out how much of the size difference observed in Kneissig et al. is due to an enrichment of ruptured MN in their lamin B1 negative population.

Reviewer #3, minor comment 5: The authors suggest that there is a linear relation between chromosome length and MN size (e.g. p.4), length and area (p. 6), and area and volume (Fig. S3A). While one can do such curve fits as presented in Figure S3A and will yield an apparent linearity because the initial slope of such curves is pretty linear, the statement is unfortunately

an oversimplification and misleading. Even though MN are not perfectly round and have the form of 'oblate spheroids', linearity is inconsistent with the underlying math (https://en.wikipedia.org/wiki/Surface-area-to-volume_ratio). To this reviewer it remains unclear why the authors would wish to suggest such linear correlations between chromosome length and MN rupture at all. One could equally well correlate MN rupture with their size/volume, which in turn correlates with chromosome length but not in a linear fashion.

The authors cite two papers to support their claim about this linear relation but both papers don't actually draw such conclusion (Eils et al., 1996; Kemeny et al., 2018). In Eils et al., Figure 6A shows a correlation between chromosome length and volume, but this is not a linear relation - the line in that Figure is linear but this is to illustrate the correlation and not linearity (e.g. compare length and volume of chromosome 1 and chromosome 22).

- Response Reviewer #3, minor comment 5: This was very informative, thank you! We have altered the text to remove the language about linearity and to focus instead on the correlation between volume and area for MN at the sizes we observed, and the general shape of MN as oblate spheroids instead of balls (page 7, line 14; page 11, line 17; figure legend 2, line 2). As the reviewer indicated, neither of these conclusions depends on an (incorrectly) assumed linear relationship. We have also removed the misleading trend line from Fig. 2A:

Reviewer #3, minor comment 6 : (1) The manuscript would benefit from an additional explanation in their discussion for the model presented in Figure 5. **(2)** It would be good if the authors better explained in the text how their current findings contribute to the different potential outcomes, i.e. aneuploidy and chromothripsis (some of it is currently hidden in the legend).

- Response Reviewer #3, minor comment 6 – part 1: This is an excellent point, and we have added additional references to Fig 5 in the discussion section to indicate when we are discussing different parts of the model. We clarified our hypothesis that higher levels of lamin B1 delay lamina gap formation in large MN, but that a different mechanism is active in MN with high gene density (discussion paragraph 1). We also reworked the second paragraph of the discussion to clarify how we think chromosome identity relates to other factors contributing to MN rupture. We hypothesize that cellular and mitotic features determine the overall likelihood of MN rupture and that MN content determines the timing of rupture for individual MN in those conditions. Thus, apparently contradictory results from other cell lines (e.g., HSA-Y in DLD=1 cells (Ly et al., 2016)) may differences in overall MN stability rather than the effect of specific chromosomes (page 11, line 1).
- Response Reviewer #3, minor comment 6 – part 2: We agree with the reviewers that this analysis needed additional explanation. We have altered the text and Fig. 5 (see below) to clarify the different DNA damage and DNA replication mechanisms potentially acting on MN that rupture in G1 versus later in the cell cycle. Although a systemic experimental analysis has not yet been

undertaken, we believe that the existing literature has sufficient evidence to support our hypothesis that early and late MN rupture have different outcomes for the micronucleated chromosome and the cell in the next interphase.

The new text now reads (page 12, line 4 to end): “Chromosome size and gene density determine whether the MN is likely to rupture in G1 or after DNA replication initiates (Fig. 5), and this could have significant effects on the consequences of MN rupture. Double-stranded DNA breaks (DSBs) in MN are thought to require DNA replication initiation and be the major type of DNA damage in MN rupturing in S and G2 or entering mitosis without rupturing (Crasta et al., 2012; Emily M. Hatch et al., 2013; Umbreit et al., 2020; Zhang et al., 2015). In contrast, rupture in G1 may promote mainly ssDNA accumulation, due to TREX-1 endonuclease activity (Mohr et al., 2021). Currently, only MN that rupture after S phase have been shown to undergo chromothripsis (Ly et al., 2019, 2016; Umbreit et al., 2020; Zhang et al., 2015), although TREX-1 has been linked to chromothripsis and hypermutation (kataegis) in other contexts (Maciejowski et al., 2020, 2015). Rupture timing also likely determines whether whole or partial chromosome aneuploidy is present in daughter cells. MN that rupture in S/G2 phase prematurely terminate DNA replication, leading to partial aneuploidy in the daughter cells, which can be exacerbated by fragment loss during chromothripsis and amplification of circularized fragments (double-minutes) (Shoshani et al., 2020; Stephens et al., 2011; Zhang et al., 2015). MN that rupture in G1 will have whole chromosome aneuploidy by G2 and likely have impaired kinetochore assembly leading to continued chromosome missegregation in the next cell cycle (Hatch et al., 2013; He et al., 2019; Soto et al., 2018b). In addition, the duration that chromatin is exposed to the cytoplasm, or the type of DNA damage, could impact whether cGAS-STING activation occurs (Guey et al., 2020; Mohr et al., 2021). It remains to be seen how differences in rupture timing, and chromosome-specific differences in transcription, replication timing, and NE assembly in MN will affect cell proliferation and immune system activation. However, our results demonstrate that identifying the chromosomes that missegregate into MN in different tissues and cancer types will be critical to understanding how MN rupture drives cancer evolution and disease pathogenesis *in vivo*.”

Reviewer #3, minor comment 7: The authors discuss an unknown factor associated with gene density and find that lamin B1 levels do not correlate. It would be interesting to see what happens to the other lamins, e.g. if they behave similar to lamin B1.

- **Response Reviewer #3, minor comment 7:** Our observation that lamin A/C recruitment for all MN is at similar to the nucleus is consistent with previous published analyses (Castro et al., 2017 and Liu et al., 2018), and therefore was not repeated for this paper. We quantified lamin B2 recruitment to MN and found that it also correlated with MN size (see below), but were unable to obtain similar data for chromosome-specific MN. This result is consistent with previously published data demonstrating that lamin B1 and B2 levels are strongly correlated in MN (Liu et al., 2018), and so instead we have included a reference to that paper in the discussion section (page 11, line 15).

Reviewer #3, minor comment 8: In the discussion, the authors state that MN have similar highly curved regions across all size ranges. However, this statement is confusing as MN size should be inversely correlated with membrane curvature. It would be good if the authors can elaborate on this statement further.

- Response Reviewer #3, minor comment 8: We have added new text to this section to minimize confusion (page 11, line 17). Average likely curvature strongly correlates with MN size, but we were struck by the presence of both high and low curvature areas on all MN regardless of size. If low curvature is sufficient for lamin B1 association with membrane, we would expect patches of lamin B1 to be present even in small MN. However, we see no evidence of lamin B1 in these areas, leading us to conclude that curvature is not the only factor contributing to defective protein recruitment and lamina disorganization in small MN.

Reviewer #3, minor comment 9: For Figure 1D, it remains unclear what the given p value actually means, and why and how the authors used the Chi-squared test on that data. This needs to be better explained.

- Response Reviewer #3, minor comment 9: We used the Chi-square test to reject the null hypothesis that all single chromosome MN have the same rupture frequency. Obtaining a significant p-value for the Chi-square family test also allowed us to calculate whether differences in rupture frequency were significant between pairs or groups of chromosomes (e.g. Fig. 1H). We have updated the materials and methods section “Statistics” to clarify this (page 17, line 23).

October 26, 2021

RE: Life Science Alliance Manuscript #LSA-2021-01210-TR

Dr. Emily M Hatch
Fred Hutchinson Cancer Research Center
Divisions of Basic Sciences and Human Biology
1100 Fairview Ave
A2-025
Seattle, WA 98109

Dear Dr. Hatch,

Thank you for submitting your revised manuscript entitled "Chromosome length and gene density contribute to micronuclear membrane stability". We would be happy to publish your paper in Life Science Alliance pending final revisions necessary to meet our formatting guidelines.

- please add an Author Contributions section to your main manuscript text
- please add a conflict of interest statement to your main manuscript text
- please consult our manuscript preparation guidelines <https://www.life-science-alliance.org/manuscript-prep> and make sure your manuscript sections are in the correct order
- please use the [10 author names, et al.] format in your references (i.e. limit the author names to the first 10)
- please add your main and supplementary figure legends to the main manuscript text after the references section
- all figure legends should only appear in the main manuscript file

A. FINAL FILES:

B. MANUSCRIPT ORGANIZATION AND FORMATTING:

Sincerely,

November 3, 2021

RE: Life Science Alliance Manuscript #LSA-2021-01210-TRR

Dr. Emily M Hatch
Fred Hutchinson Cancer Research Center
Divisions of Basic Sciences and Human Biology
1100 Fairview Ave
A2-025
Seattle, WA 98109

Dear Dr. Hatch,

Thank you for submitting your Research Article entitled "Chromosome length and gene density contribute to micronuclear membrane stability". It is a pleasure to let you know that your manuscript is now accepted for publication in Life Science Alliance. Congratulations on this interesting work.

DISTRIBUTION OF MATERIALS:

Again, congratulations on a very nice paper. I hope you found the review process to be constructive and are pleased with how the manuscript was handled editorially. We look forward to future exciting submissions from your lab.

Sincerely,
